# Efficient and versatile formation of glycosidic bonds via catalytic strain-release glycosylation with glycosyl *ortho*−2,2-dimethoxycarbonylcyclopropylbenzoate donors

Han Ding [1], Jian Lyu [1], Xiao-Lin Zhang [1], Xiong Xiao [2] ✉ & Xue-Wei Liu [1] ✉

Catalytic glycosylation is a vital transformation in synthetic carbohydrate chemistry due to its ability to expedite the large-scale oligosaccharide synthesis for glycobiology studies with the consumption of minimal amounts of promoters. Herein we introduce a facile and efficient catalytic glycosylation employing glycosyl *ortho*−2,2-dimethoxycarbonylcyclopropylbenzoates (CCBz) promoted by a readily accessible and non-toxic Sc(III) catalyst system. The glycosylation reaction involves a novel activation mode of glycosyl esters driven by the ring-strain release of an intramolecularly incorporated donor-acceptor cyclopropane (DAC). The versatile glycosyl CCBz donor enables highly efficient construction of *O*-, *S*-, and *N*-glycosidic bonds under mild conditions, as exemplified by the convenient preparation of the synthetically challenging chitooligosaccharide derivatives. Of note, a gram-scale synthesis of tetrasaccharide corresponding to Lipid IV with modifiable handles is achieved using the catalytic strain-release glycosylation. These attractive features promise this donor to be the prototype for developing next generation of catalytic glycosylation.

An increasing realization of the significant roles of carbohydrates in various biological processes including cell growth, cell-cell adhesion, fertilization and immune response[1,2] in the past decades has translated into a surging demand for glycans and glycoconjugates with well-defined structures in sufficient quantities for deciphering their biological functions. To achieve chemical preparation of natural oligosaccharides and glycoconjugates difficult to isolate on large scale from biological sources and artificially designed carbohydrate-based pharmaceutics[3], numerous glycosylation methods have been developed for constructing various glycosidic linkages.

Among a myriad of glycosylation protocols, catalytic glycosylation reaction is one of the most exciting fields in synthetic carbohydrate chemistry, because it offers unique opportunities for preparing large quantities of glycans and glycoconjugates without consuming a stoichiometric amount of promoters, typically strong electrophiles or Lewis acid[4,5] which could not only incur undesired side reactions, but also induce environmental concerns. While efficient catalytic activation methodologies under mild conditions have been developed for glycosyl donors including glycosyl fluorides[6–8], imidates[9,10], epoxides[11,12] and phosphates[13–15], the activated glycosyl donors are

[1]School of Chemistry, Chemical Engineering and Biotechnology, Nanyang Technological University, 21 Nanyang Link, Singapore 637371, Singapore. [2]School of Chemistry and Chemical Engineering, Northwestern Polytechnical University (NPU), Xi'an 710072, P.R. China. ✉e-mail: xiongxiao@nwpu.edu.cn; xuewei@ntu.edu.sg

generally highly reactive and sensitive to moisture, thus most of them should be prepared freshly prior to the glycosylation stage. A classical family of glycosyl donors with good stabilities is thioglycosides. Although multiple variations of thioglycoside have been developed to realize efficient catalytic glycosylation reactions[16–19], there is still a lot of space for the development of catalytic activation of classic thioglycoside. From a chronology with the cornerstone of catalytically activable glycosyl donors, we have identified a growing demand for ambiently stable donors that can be activated with catalytic amounts of promoters (Fig. 1). Inspired by these considerations, we endeavored to design a bench-top stable glycosyl donor that can be selectively activated by a specific catalyst in an orthogonal manner, utilizing a novel activation mode.

Glycosyl esters, easily accessible from anomeric hemiacetals and benchtop stable, are another type of promising glycosyl donors for oligosaccharide synthesis. However, simple glycosyl esters like

glycosyl acetates and benzoates have mediocre reactivities and glycosylation reactions employing these donors have narrow acceptor scopes and require the addition of superstoichiometric amounts of promoters[4,20]. Donors with certain modified ester leaving groups such as glycosyl pentenoates[21], glycosyl *ortho*-(1-phenylninyl)benzoates[22], glycosyl heteroaromatic carboxylate esters[23–25], and glycosyl allenoates[26] show enhanced reactivities and expanded acceptor scopes, while few catalytic activation methods for glycosyl ester donors are available to date.

Up to present, two reaction modes for catalytically activating glycosyl esters are developed, namely, activation through the formation of a covalent bond or a non-covalent interaction between the activation site of the donor and catalyst. For the former, very recently, Li and Kancharla independently reported superacid Tf$_2$NH-catalyzed *O*-, *C*-, and *S*-glycosylation reactions of glycosyl esters based on alkene chemistry[27,28]. These reactions provide the chances for the catalytic

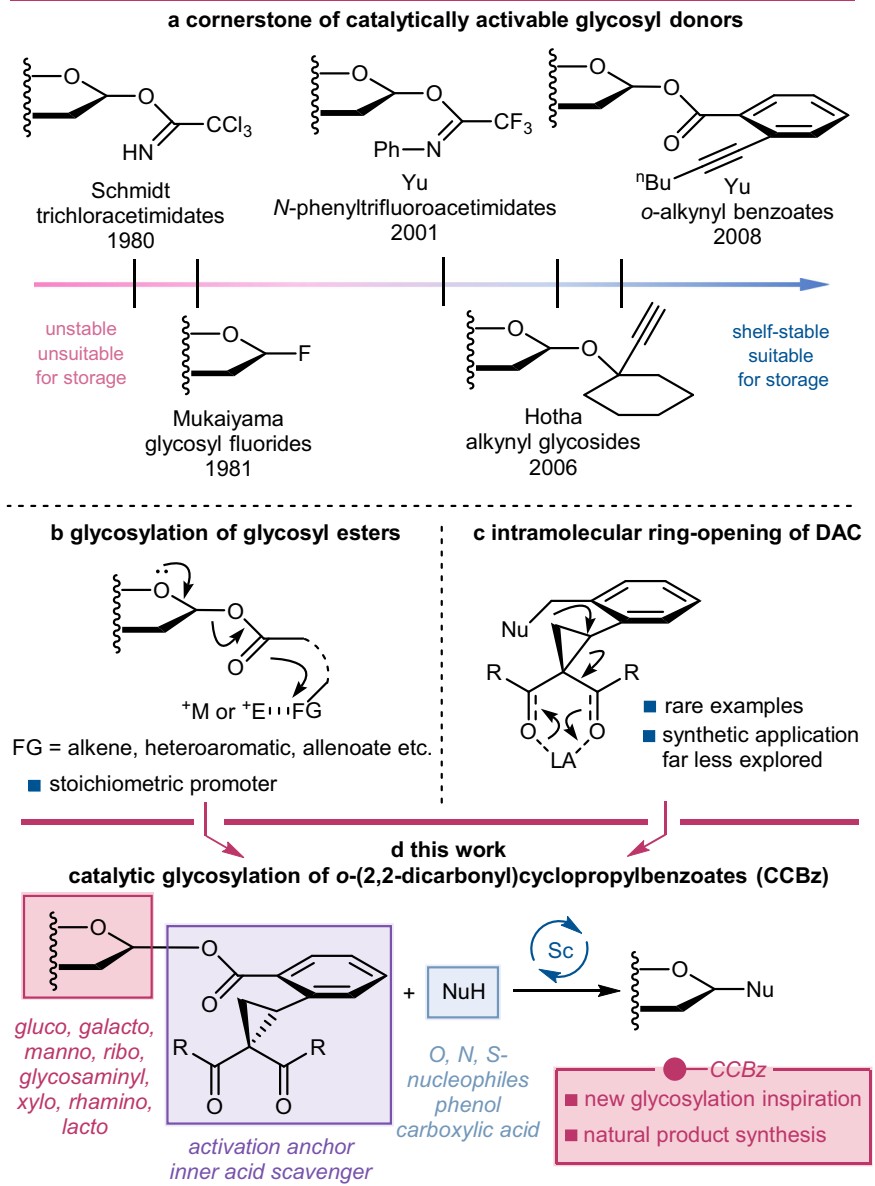

**Fig. 1 | Major catalytic glycosylation reactions available to date. a** The cornerstone glycosyl donors for catalytic glycosylation in chronological order. **b** Stoichiometrically remote activation of glycosyl ester type donor with strong electrophiles and transition metals. **c** Intramolecular ring-opening of donor-acceptor cyclopropane (DAC). **d** Our report on catalytic glycosylation with glycosyl *ortho*-(2,2-dicarbonyl)cyclopropylbenzoate (CCBz). Ph Phenyl, ⁿBu *n*-butyl, M Transition metals, E Electrophiles, FG Functional group, R Functional group, Nu Nucleophile, LA Lewis acid.

formation of the glycosidic bonds. However, the strongly acidic conditions of the reactions potentially prohibit their application to acceptors with acid-labile protecting groups like acetonide or benzylidene. Moreover, the superacid-catalyzed glycosylation reactions suffer from a significant aglycone transfer side reaction when employing thioglycosides, extensively used for chemical glycosylation, as acceptors.

On the other hand, activating a functionality at the distant side of the leaving group within the glycosyl ester through a weak interaction between the catalyst and activable anchor can minimize the side reactions like catalyst deactivation by undesired interaction between catalyst and donor or acceptor equipped with multiple delicate functionalities, ensuring the glycosylation reaction to proceed smoothly in milder environment. The seminal work of gold(I)-catalyzed glycosylation with glycosyl *ortho*-alkynylbenzoates by Yu et al.[29] provides an elegant and versatile approach for synthesizing structurally diverse natural oligosaccharides and glycoconjugates of biological relevances[30,31]. Yu's reaction activates the donor through a weak interaction between the metal catalyst and the alkyne leaving group and suppresses the aglycone transfer side reactions which plague the traditional glycosylation reactions with covalent activation modes. Inspired by this elegant activation mode, a series of catalytic glycosylation reactions have been developed recently for efficient construction of *O*-[32–35] and *N*-glycosidic linkages[36–38]. The glycosylation reactions employing glycosyl alkynes catalyzed by expensive gold, silver or highly toxic mercury catalysts thus remain the prevailing strategy for catalytic glycosylation with weak-interaction-mediated activation[39–41].

Molecular structures with strained rings have tunable reactivities due to their inherent trend to release the ring strain, making them divergent scaffolds for organic synthesis[42–44], materials science[45] and bioconjugation[46,47]. Strained ring-containing compounds including the donor-acceptor cyclopropanes (DACs) /cyclobutanes (DABs)[48,49] and a series of "spring-loaded" reagents[50] have high π characters and undergo homologous conjugate addition with various nucleophiles. Although preparation and investigation of DACs have been reported 30 years ago, only until recently DACs were incorporated into the toolbox of organic synthesis. With significant ring-strain, DAC agents readily undergo conjugate additions with various *C*-, *N*-, *O*-, *S*-, and *Se*-nucleophiles in either intermolecular or intramolecular fashion, initiated by a non-covalent chelation process. The intermolecular conjugate addition has been exploited in the synthesis of natural products[51–53], while the synthetic potential of the intramolecular version has barely been explored[54]. In addition, while catalytic glycosylation reactions involving cyclopropane-fused glycosyl donors have successfully demonstrated excellent control of anomeric selectivity to produce unnatural glycosides[55,56], an effective approach to obtain natural sugar derivatives via catalytic strain-release-driven glycosylation remains elusive.

Recognizing the diverse reaction profiles of the DACs under Lewis acidic conditions, we envisioned that by introducing a DAC structure into the glycosyl ester donors, we could construct a new genre of glycosyl donors with an activation mode mediated by non-covalent interactions. Herein, we report a rationally designed glycosyl donor with an intramolecularly incorporated DAC featuring a dual-functional anchor: The metallophilic 1,3-dicarbonyl group as the activation site, and the ensuing enolate as an acid scavenger. The glycosylation reaction proceeds smoothly under mild conditions and applies to broad donor and acceptor scopes, enabling the construction of challenging glycosidic bonds, as demonstrated by the facile assembly of chitooligosaccharide derivatives. This glycosylation promises to provide access to next generations of carbohydrate-based therapeutics and inspire following studies of glycosylation methodologies to tap into recent advancements in synthetic organic chemistry.

## Results
### Design plan
The proposed mechanism of the putative activation of glycosyl *ortho*−2,2-dicarbonylcyclopropylbenzoates **A** is outlined in Fig. 2[57]. The chelatable Lewis acid can selectively chelate with two carbonyl groups within the aglycone to activate the C-C bond in the cyclopropyl group. The nucleophilic attack of the carbonyl group in the benzoyl group can open the cyclopropyl group to form the enolate, accompanied by the ready cleavage of the anomeric C-O bond to generate the oxocarbenium **C**. **C** could be engaged in the glycosylation reaction to couple with a nucleophile to give the protonated product **E**. The deprotonation of **E** by enolate **D** finally generates lactone **F** as leaving group and desired product **G** to close the catalytic cycle by releasing the Lewis acid catalyst for the next cycle of glycosylation. It is worthwhile to indicate that enolate **F** not only takes the role of the leaving group merely but can scavenge the free proton released by the acceptor to maintain the reaction system nearly neutral. This meticulously designed glycosylation reaction thus exhibits the versatility of the dual-tasked DAC.

To reduce the idea into reality, however, several concerns should be taken into consideration from the mechanistic view. First, even though the nucleophilic ring-opening of DAC by an intramolecular nucleophile is known, most of those reactions require harsh reaction conditions such as stoichiometric strong Lewis acid promoters and very high temperature up to 120 °C[58]. Those drawbacks will cause the decomposition of oligosaccharides and glycoconjugates bearing delicate functional groups. Another is that the report using carboxyl acid as the nucleophile to open the DAC is rare. The sole example using carboxylic acids as nucleophiles was reported by Feng's group, albeit 2.5 equiv of benzoic acid is essential to secure the high yield at 60 °C[59]. A related issue is that we aimed to develop a glycosylation reaction with such the CCBz donor could be selectively activated over other glycosyl donors. Even though new glycosyl donors are developed each year, the conditions to activate most of them also activate other common glycosyl donors, causing unbiased glycosylation reactions. Thus, the development of orthogonal glycosyl donors plays a critical role to simplify the synthesis of oligosaccharides and glycoconjugates, for example, the one-pot oligosaccharide synthesis. Last but not least, for most glycosylation reactions with ester type donors, the 1,2-*trans* glycosyl donors are necessary for high yields due to their higher reactivity resulting from the push effect of the C-2 anchimeric group, and whether the 1,2-*cis* glycosyl CCBz could be effectively activated by Lewis acid is questionable.

### Reaction development
Our study commenced with the development of a facile and robust route toward the designed glycosyl *ortho*−2,2-dicarbonylcyclopropylbenzoate, which could be achieved conveniently by the coupling reaction between anomeric hemiacetals and *ortho*−2,2-dimethoxycarbonylcyclopropylbenzoic acid (CCBzOH, for the detailed synthesis of CCBzOH with tips and tricks, see SI) in the presence of 1-(3-dimethylaminopropyl)−3-ethylcarbodiimide hydrochloride (EDC·HCl), *N*,*N*-diisopropylethylamine (DIPEA) and 4-dimethylaminopyridine (DMAP), and various glycosyl CCBzs could be obtained in good-to-excellent yields (see SI). Of note, the CCBzOH can be prepared on a decagram scale in a single batch and all resulting donors are amorphous solids with good solubility in common organic solvents and excellent thermostability, making them suitable for storage under the ambient environment for at least several months without any indication of decomposition or hydrolysis.

Subsequently, the key strain-release glycosylation was optimized by using the coupling reaction between disarmed glucosyl CCBz **1a** and cholesterol **2a** (Table 1). A series of Lewis acids were initially tested in the presence of 5 Å molecular sieve (MS) in a 0.05 M solution of **1a** in 1,2-dichloroethane (DCE) at room temperature for 2 to 5 h. To our delight, 0.1 equiv of Sc(OTf)$_3$ showed very high affinity to the carbonyl

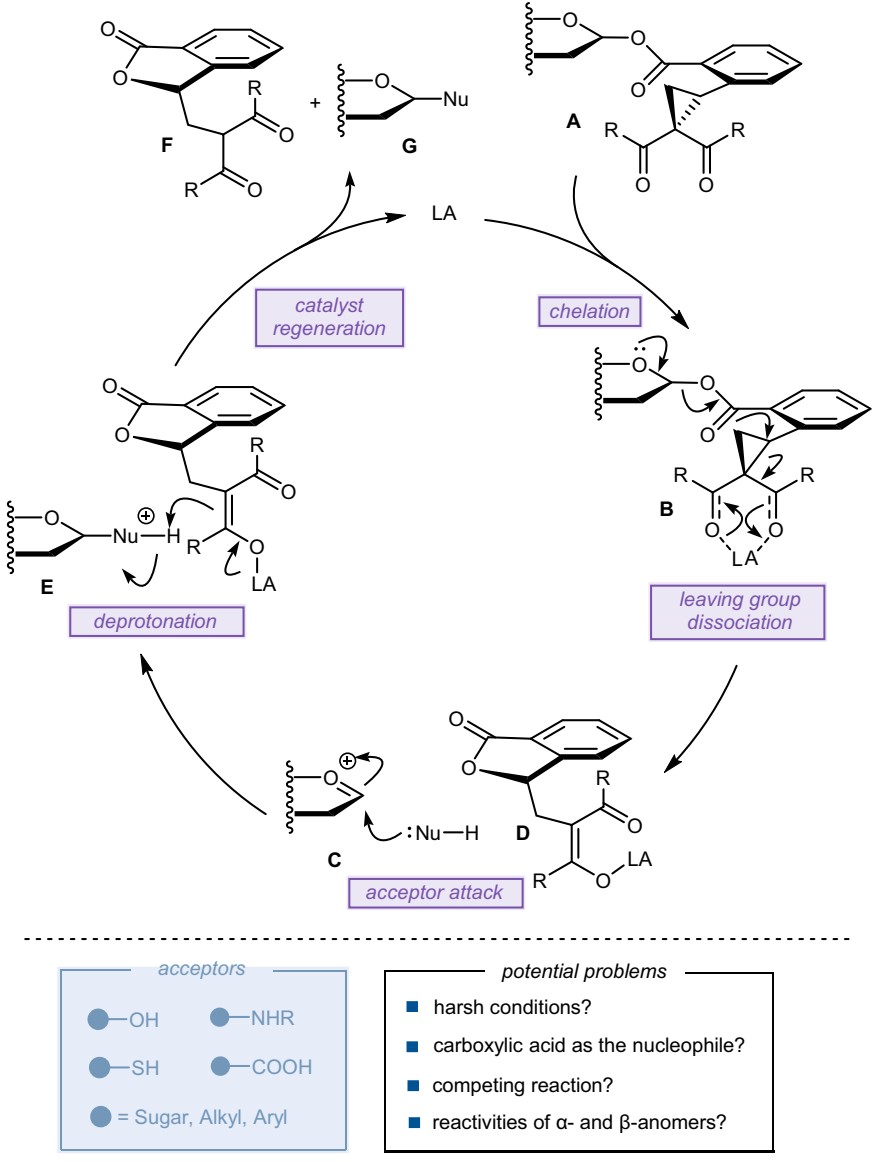

**Fig. 2 | Reaction design.** The working hypothesis of the present work: the chelation of the Lewis acid with the dicarbonyl group of DAC can initiate the ring-strain-release-induced glycosylation reaction followed by the proton scavenging by released enolate. R Functional group, LA Lewis acid, Nu Nucleophile.

group of the CCBz donor and the desired product **3a** and departing product **4** were separated in excellent yields of 96% and 99%, respectively without any acceptor inserting into the DAC detected, which attested to our proposal (Table 1, entry 1). The satisfying result indicated that both α and β isomers of **1a** could be activated to participate in the glycosylation. Without the Lewis acid, however, no reaction occurred and the **1a** could be recovered in almost quantitative yield (Table 1, entry 2). Other Lewis acids capable of activating DAC like Bi(OTf)$_3$, Zn(OTf)$_2$, B(C$_6$F$_5$)$_3$ and a combination of Ca(OTf)$_2$/$^n$Bu$_4$NPF$_6$ showed inferior yields or no conversion of the starting material, while TfOH gave complicated result (Table 1, entries 3-5).

Given the fact that the glycosyl CCBz could be selectively activated by Sc(OTf)$_3$, this glycosylation might be a new type of donor that could be incorporated into catalytically orthogonal one-pot glycosylation. Thus, several representative catalysts for different glycosyl donors were further tested. The results suggested that

TMSOTf (trimethylsilyl trifluoromethanesulfonate, glycosyl trichloroacetimidate and glycosyl *N*-phenyltrifluoroacetimidates)[9,10] and Ph$_3$PAuNTf$_2$ (glycosyl *ortho*-alkynylbenzoates)[36–38] could only give trace amount of **3a** or even no desired product, demonstrating that glycosyl CCBz could be specifically activated by the chelatable transition metal Lewis acid catalyst, revealing a family of orthogonal glycosyl ester type donors (Table 1, entries 6-7). Thus, the current donor potentially premises a new link for one-pot orthogonal glycosylation.

In keeping with other acid-catalyzed glycosylation reactions[27,60], the molecular sieve (MS) played a significant role in our glycosylation. When the more basic 4 Å MS was used as the dehydrating agent, a dramatic decrease in yield was observed (35%, Table 1, entry 8). By screening the commonly used organic solvents, we found that this glycosylation performed well in all types of organic solvents and CH$_2$Cl$_2$ give the best yield of 99% (Table 1, entries 9-10). Based on the experiments, as shown in Table 1, entry 8, the condition composed of

**Table 1 | Reaction development and the control experiment[a]**

| Entry | Derivation from standard condition | Yields of 3a (%) | Yields of recovered donor (%) | Yields of 4 (%) |
|---|---|---|---|---|
| 1 | None | 96 | - | 99 |
| 2 | No Sc(OTf)$_3$ | 0 | >95 | 0 |
| 3 | Bi(OTf)$_3$ as the catalyst | 65 | 40 | 58 |
| 4 | Zn(OTf)$_2$ or Ca(OTf)$_2$/$^n$Bu$_4$NPF$_6$ or B(C$_6$F$_5$)$_3$ as catalyst | 0 | >95 | 0 |
| 5 | TfOH as catast | complex mixture | - | NA |
| 6 | TMSOTf as catalyst | <5 | >95 | <5 |
| 7 | Ph$_3$PAuNTf$_2$ as catalyst | 0 | >95 | 0 |
| 8 | 4 Å MS instead of 5 Å MS | 33 | 68 | 30 |
| 9 | CH$_2$Cl$_2$ as the solvent | 99 | - | 99 |
| 10 | PhCH$_3$ or PhCF$_3$ or Et$_2$O as solvent | 96–97 | - | >95 |
| 11[b] | glucose pentabenzoate 5 as donor | trace | >95 | NA |

[a]Unless otherwise specified, all reactions were performed with 1.2 equiv of **1a**, 1.0 equiv of **2a** (0.05 mmol) in the presence of the catalyst (0.1 equiv) and 5 Å MS in corresponding solvent (0.05 M, 1 mL) for 2–5 h at room temperature. The yield for **3a** was based on **2a**, and the yields for **4** and the recovered donor were based on the **1a**. [b]CH$_2$Cl$_2$ was used as the solvent. *Bz* Benzoyl, *Me* Methyl, *Tf* Trifluoromethanesulfonyl, *DCE* 1,2-dichloroethane, $^n$*Bu* n-Butyl, *TMS* Trimethylsilyl, *Ph* Phenyl, *MS* Molecular sieve, *Et* Ethyl, *NA* Not applicable.

glycosyl CCBz donor (1.2 equiv), acceptor (1.0 equiv), Sc(OTf)$_3$ (0.1 equiv), 5 Å MS and CH$_2$Cl$_2$ (0.05 M of acceptor) as the solvent was selected for our study on catalytic strain-release glycosylation with glycosyl CCBz. Finally, a preliminary control experiment was conducted to corroborate our proposal (Table 1, entry 11). When glucose pentabenzoate **5** was exposed to the optimal condition of our catalytic strain-release glycosylation, only a trace amount of the desired product was formed, indicating that unlike the traditional mechanism of glycosylation reactions with glycosyl esters, the catalyst activated the DAC site by chelating with two carbonyl groups, which offered a new activation mode of ester type glycosyl donors.

## Substrate scope studies

With optimal condition for our strain-release glycosylation in hand, we then turned out to test the generality of the established glycosylation reaction. Thus structurally diverse acceptors **2b-s** were coupled with **1a** and the results were outlined in Fig. 3. From these results, it is obvious that aliphatic alcohols with primary, secondary, and tertiary hydroxyl groups, sugar alcohols, carboxylic acid, phenol and heteroatom nucleophiles were all competent acceptors, affording the desired glycosides and disaccharides in good-to-excellent yields with complete β-selectivities with resort to the anchimeric assistance by *O*−2 benzoyl group. Of note, for most aliphatic and sugar alcohols, over 95% yields were obtained regardless of the structures of the acceptors, demonstrating the high efficiency of the well-established protocol. A distinguishing feature of the strain-release glycosylation was that the glycosyl CCBz could be selectively activated over the thioglycoside to afford the disaccharide thioglycosyl donor **3 g** ready for the next glycosylation, which can streamline the oligosaccharide synthesis. More importantly, it is interesting to suggest that we didn't observe the aglycone transfer of the anomeric phenylthiol group, a very common side reaction for the glycosylation of thioglycoside acceptor, which set a solid foundation for orthogonal glycosylation based on the CCBz

strategy. To our delight, the acid-sensitive acetonide and benzylidene group were well-tolerated to generate the products **3h-j** in excellent yields without any detection of the deprotection of the acetonide and benzylidene. The *C*2-OH of the mannoside **2k** and *C*4-OH of the methyl glucuronate **2 l** are usually considered challenging nucleophiles due to the poor nucleophilicity resulting from the electro-withdrawing effect of the *C*−2 axial-oriented bond of mannosides and the *C*−5 carboxyl group of glucuronates, respectively. However, under our conditions, the two glycosylation reactions proceeded smoothly to deliver the corresponding *O*−2 and *O*−4 linked disaccharides **3k** and **3 l** in 89% and 91% under very mild conditions. *N*-Hydroxylphthalimide **2 m** and *ortho*-iodobenzoic acid **2n** were good coupling partners to glycosylate with **1a** to afford the intended glycosides, which are latent forms of *N*-(glycosyloxy)acetamides[61] and *ortho*-alkynylbenzoates, showcasing the versatility and flexibility of the strain-release glycosylation. It should also be noted that the previous preparation of aminooxy glycoside **3 m** entailed unstable glycosyl halides in the presence of stoichiometric silver salt or propargyl 1,2-orthoesters with a gold catalyst to give the products in only moderate yields[61,62]. Our facile synthesis of **3 m** gave an example of the catalytically feasible method using cheap rare earth metal as the catalyst, which could supply potentially easier access to various aminooxy glycosides of biological interests like calicheamicin[63] and artificial glycopeptides[64]. Finally, the *S*-acceptors were also viable for the construction of *S*-linked glycosides, as exemplified by the efficient coupling reactions of donor **1a** with aromatic thiol **2q** and aliphatic thiol **2r**, respectively. It should be noted that in all cases of successful glycosylation reactions employing heteroatom nucleophiles, the direct ring-opening of DAC moiety by acceptors was not observed, denoting the intramolecular cyclization can be kinetically favorable even in the presence of a heteroatom nucleophile. However, it was disappointing that when galactose-derived anomeric thiol **2 s** was subjected to our optimal condition, no reaction occurred. Currently, the reason for the unpleasant consequence was unknown

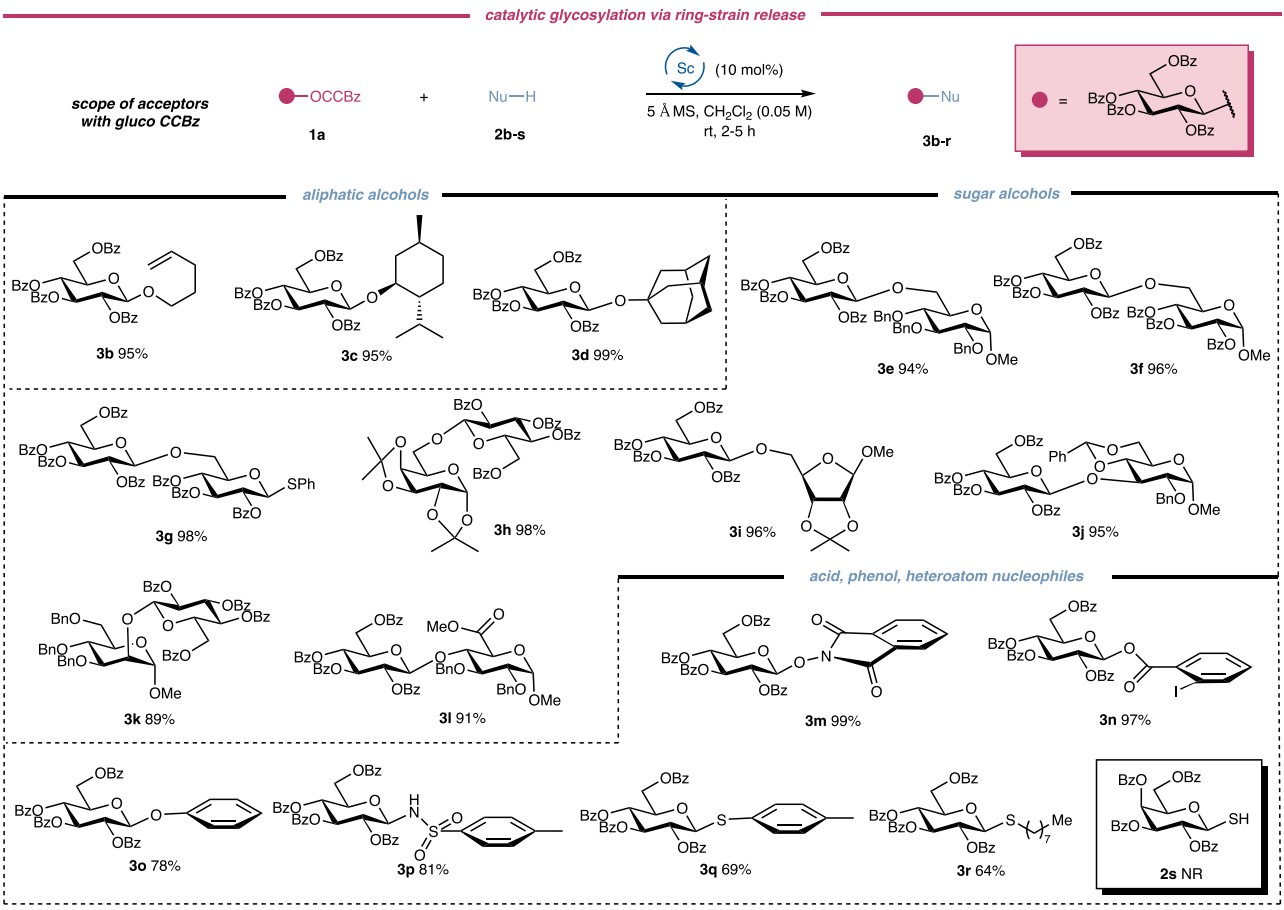

**Fig. 3 | Acceptor scope of catalytic strain-release glycosylation with glucosyl CCBz donor.** Acceptors ranged from aliphatic alcohols, sugar alcohols with primary, secondary and tertiary hydroxyl groups, benzoic acid derivative, phenol, sulfonamide and thiols are compatible coupling partners in strain-release glycosylation. Unless otherwise stated, reported yields are for isolated and purified products. For alcoholic nucleophiles, 1.2 equiv of donor and 1.0 equiv of acceptor were used for glycosylation reaction; for acid and heteroatom nucleophiles, 1.0 equiv of donor and 1.5 equiv of acceptor were used for the glycosylation reaction. CCBz, ortho−2,2-dimethoxycarbonylcyclopropylbenzoyl; rt, room temperature; Nu, nucleophile; Bz, benzoyl; Bn, benzyl; Ph, pheny; Me, methyl.

and we are still working on the synthesis of 1,1'-thiosaccharide congeners by strain-release glycosylation because of their important roles as drug candidates[65–67].

The generality of the current glycosylation reaction was further tested by glycosylating a series of CCBz donors with three types of acceptors, namely, the aliphatic adamantanol (Nu[1]), 6-OH of thioglycoside as a sugar nucleophile (Nu[2]) and the ortho-iodobenzoic acid as carboxylic acid acceptor (Nu[3]). As demonstrated in Fig. 4, D-galactosyl, D-mannosyl, D-glucosaminyl, D-xylopyranosyl, L-rhamnopyranosyl, D-ribofuranosyl, lactosyl CCBzs 1b-h gave the desired 21 oligosaccharides and glycosides in good-to-excellent yields, showing the feasibility of CCBz strategy for the synthesis of pyranosides, furanosides, deoxy sugars, amino sugars and L-sugars. Note that the aglycon transfer of thioglycoside acceptor was not observed for most CCBz donors, implying the orthogonal ability of the current glycosylation reaction. The successful glycosylation with D-glucosaminyl donor 1d and D-ribofuranosyl donor 1g indicated that the acetyl group is also a viable protecting group for our glycosylation and no orthoester was isolated from all the reactions. The disaccharide lactosyl CCBz 1h could be glycosylated with the three types of acceptors in excellent yields, showing the potential of this glycosylation for the synthesis of bigger oligosaccharides.

## Divergent synthesis of chitooligosaccharides
Chitooligosaccharides (COS) is recently gathered much attention because they share similar functions with chitin and chitosan

meanwhile ameliorate the drawbacks of the two polymers resulting from the high molecular weight, poor solubility and high viscosity of chitosan solutions[68]. Besides, the COS derivatives modified at a different position on the backbones of the COS with well-defined structures have more interesting properties, as exemplified by TMG-chitotriomycin[69], Nod factors[70,71] and Myc factors[72] which demonstrate diverse bioactivities. Lipid II and Lipid IV[73–75], key donors in the biosynthesis of peptidoglycans, are the decisive substrates for bacterial cell wall biosynthesis by bacterial transglycosylase (TGase), which contains conserved catalytic residue and serves as a promising antibiotic target especially for drug-resistant bacteria like vancomycin-resistant *Enterococci* (VRE) and methicillin-resistant *Staphylococcus aureus* (MRSA)[76–78].

However, the uncertainty belies the huge achievements toward deciphering the critical functions of COS derivatives mainly due to the acute scarcity of natural reserves for such substrates. The chemical synthesis unlocks valuable access to COS derivatives on a considerable scale with precise structures[79]. Many glycosyl donors have been successfully applied to the synthesis of COS and congeners of biological interests, including glycosyl halide[80,81], glycosyl acetimidate[82–84], thioglycosides[81,85–88], *n*-pentenyl glycosides[89], glycosyl *ortho*-alkynylbenzoate[90,91], glycosyl sulfoxide[92], glycosyl oxazoline[93] and glycosyl *ortho*-(1-phenylvinyl)benzoate[94]. Among these elegant protocols, only Yu's glycosyl *ortho*-alkynylbenzoate, however, can serve as the stable while catalytically activable glycosyl donor. To demonstrate the feasibility of our glycosyl CCBz as a stable but efficient glycosylating agent, we decided to conduct a diverse assembly of

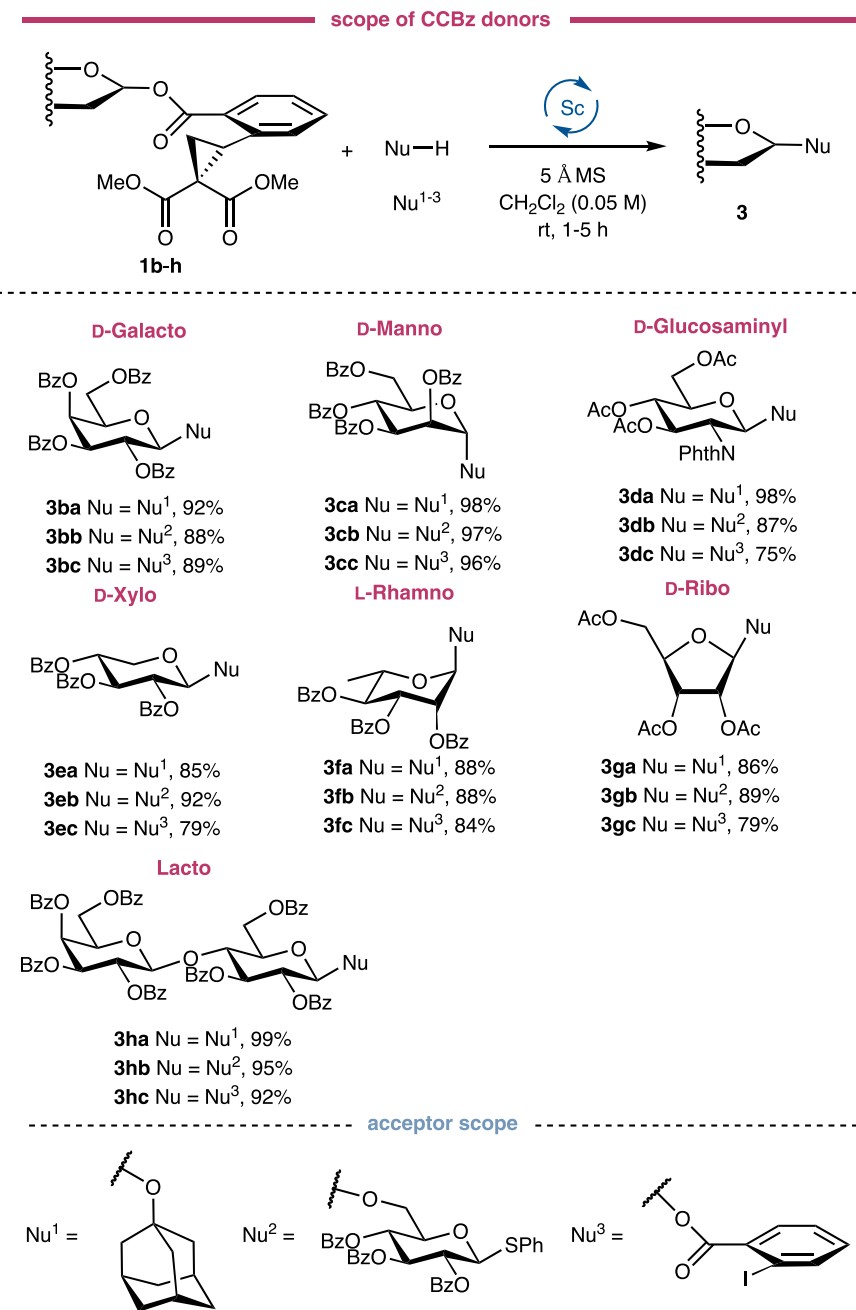

**Fig. 4 | Donor scope with different acceptors.** Glycosyl CCBzs derived from different monosaccharide- or oligosaccharide-based hemiacetals are competent coupling partners for strain-release glycosylation. Unless otherwise noted, all reported yields are isolated and purified products. For glycosylation reactions of Nu¹ and Nu², 1.2 equiv of donor and 1.0 equiv of acceptor were used; for glycosylation reactions of **Nu³**, 1.0 equiv of donor and 1.5 equiv of acceptor were used. rt, room temperature; Nu, nucleophile; Bz, benzoyl; Ac, acetyl; Ph, phenyl; PhthN, phthalimido.

chitooligosaccharides, including the gram-scale preparation of tetrasaccharide corresponding to Lipid IV. Structurally, the chitooligosaccharides are all linked by a β-D-(1→4)-glucosaminic bond. Owing to the inherently poor nucleophilicity of *C*4-OH of glucosaminosides, this acceptor is recognized as one of the most difficult sugar acceptors. Considering the *C*4-OH of methyl glucouronate, another generally acknowledged poor nucleophile, is well-accommodated to our strain-release glycosylation, the CCBz protocol should be suitable for the assembly of these COS analogues.

Our synthesis commenced with the installation of the CCBz group on the hemiacetal **6** and the desired CCBz donor **7** was obtained as the sole β-isomer in the yield of 80% (Fig. 5). To our delight, when arming **7** was employed, the strain-release glycosylation of acceptor **8** produced the desired disaccharide **9** in an excellent yield of 96% and with complete β-selectivity with resort to *N*−2 phthalimido group in the presence of only 5 mol% of Sc(OTf)₃, implying that not only such poor nucleophile is compatible with our glycosylation reaction, but also the armed glycosyl donor equipped with benzyl group is viable for the oligosaccharide synthesis using our protocol. Note that this disaccharide is also the common intermediate for the synthesis of corepentasaccharide of *N*-glycans. Deacetylation in the presence of NaOMe afforded the disaccharide acceptor **10**, which was readily glycosylated with **7** to generate the trisaccharide **11** in two steps. Concomitant removal of phthalimido groups and the acetyl group by

**Fig. 5 | Divergent assembly of chitooligosaccharides.** The synthesis of chitooligosaccharide was performed by a (1 + 1 + 1 + 1) glycosylation strategy. Ac Acetyl, Me Methyl, Bn Benzyl, PhthN Phthalimido, R Functional group, DMAP 4-dimethylaminopyridine, EDC·HCl *N*-Ethyl-*N*′-(3-dimethylaminopropyl)carbodiimide hydrochloride, DIPEA *N,N*-diisopropylethylamine, NIS *N*-Iodosuccinimide, TMSOTf trimethylsilyl trifluoromethanesulfonate, THF Tetrahydrofuran, Cbz Benzyloxycarbonyl, Ph Phenyl, CCBz, *ortho*−2,2-dimethoxycarbonylcyclopropylbenzoyl.

ethylenediamine, followed by acetylation of free amine groups and the hydroxyl group gave the natural form of COS derivative **12** in 84% in two steps. Selective deacetylation of the acetate in the presence of three acetylamino groups gave the trisaccharide acceptor with an excellent yield of 93%. Exposing the trisaccharide acceptor to *N*-iodosuccinimide (NIS)/TMSOTf promoted glycosylation using glycosyl selenide **14** as the donor completed the assembly of the tetrasaccharide **16**, albeit in the inferior yield, further showing the poor nucleophilicity of 4-OH of glucosaminoside using traditional glycosyl donors. We also prepared glycosyl CCBz donor **15** from glycosyl selenide **14** in two steps to test the viability of glycosyl CCBz in the tetrasaccharide assembly (For the experimental details, please see SI). To our delight, this glycosylation reaction gave an improved yield of 58% under established condition. The enhanced reactivity and convenient operation further demonstrate the usability of glycosyl CCBz. The global deprotection of benzyl groups and benzyloxycarbonyl (CBz) group by hydrogenolysis in a mixed solvent of *i*PrOH/THF/H₂O led to the free tetrasaccharide **17** in 85% yield after purification. The transformation of **17** into bioactive COS derivatives could be achieved in a single step according to the literature[93].

With a successful route toward COS natural products established based on the CCBz strategy, we then decided to make the tetrasaccharide ready for synthesizing Lipid IV on a gram scale (Fig. 6). The hemiacetal **18** was coupled with CCBzOH with the assistance of EDC·HCl to give donor **19** in the yield of 76% with β-only selectivity, which was activated over thioglycoside acceptor **20** to afford the Lipid II disaccharide **21** in 72%. It should be noted that the lower yield was attributed to the unavoidable elimination of the donor when the phthalimido group was used as the protecting group of amine instead

of the aglycon transfer side reaction, which was supported by the recovery of unreacted acceptor. The disaccharide was obtained on a 5.19-gram scale, which was then exposed to fluoride-mediated desilylation or anomeric hydrolysis and esterification sequence to generate Lipid II acceptor **22** and Lipid II donor **23** in 86% yield and 86% yield in two steps, respectively. To our delight, the coupling reaction between **22** and **23** proceeded smoothly and succeeded the assembly of Lipid IV tetrasaccharide **24** in a satisfying yield of 68% on a 1.22-gram scale in a single batch. Following the literature[88], this tetrasaccharide could facilely be transformed into Lipid IV. Likewise, only a trace amount of aglycon-transferred side product was observed, further demonstrating the feasibility of our strain-release glycosylation in the synthesis of complex oligosaccharides using thioglycoside as the acceptor. Several orthogonally modifiable sites on the tetrasaccharide could be selectively furnished to obtain a series of Lipid IV derivatives that hold the potential as bacterial TGases-targeting antimicrobial agents. We believe the scalable route toward Lipid IV tetrasaccharide based on strain-release glycosylation will finally contribute to the development of the next generation of antibiotics.

## Discussion

In conclusion, we have developed an easy to conduct and efficient glycosylation reaction employing bench-stable glycosyl CCBz as the donor, which involves a ring-strain release activation mode mediated by non-covalent interactions. This glycosylation reaction proceeds under mild conditions and tolerates broad donor and acceptor scopes, enabling effective *O*-, *S*-, and *N*-glycosylation, including glycosylation of thioglycoside acceptors with minimal aglycone transfer side reaction. Importantly, the glycosyl CCBz donor is completely inert under

**Fig. 6 | Scalable assembly of tetrasaccharide corresponding to Lipid IV.** The formal synthesis of Lipid IV was performed by a [(1 + 1) + 2] glycosylation strategy. Bn Benzyl, TBS *tert*-butyldimethylsilyl, PhthN phthalimido, Bz Benzoyl, NBS *N*- bromosuccinimide, DMAP 4-dimethylaminopyridine, EDC·HCl *N*-Ethyl-*N*'-(3-dime-thylaminopropyl)carbodiimide hydrochloride, DIPEA *N,N*-diisopropylethylamine, Tol tolyl, Ac acetyl, Ala Alanine, Glu Glutamic acid, Lys Lysine.

various established conditions of previously reported catalytic glycosylation reactions and can potentially be used in one-pot synthesis of oligosaccharides. As demonstrated by the streamlined assembly of chitooligosaccharide and Lipid IV tetrasaccharide, this glycosyl donor can be a potential candidate for a new genre of catalytically activable glycosyl donors to synthesize complex oligosaccharides.

## Methods

### General procedure for catalytic strain-release glycosylation

To an oven-dried 5 mL vial containing freshly activated 5 Å MS (100 mg) and a dry magnetic stir bar were added sequentially glycosyl CCBz donor (0.06 mmol, 1.2 equiv for alcoholic nucleophiles or 0.05 mmol, 1.0 equiv for acid and heteroatom nucleophiles), acceptors (0.05 mmol, 1.0 equiv for alcoholic nucleophiles or 0.075 mmol, 1.5 equiv for acid and heteroatom nucleophiles). Anhydrous $CH_2Cl_2$ (1.0 mL) was added to dissolve the starting materials and the mixture was stirred at room temperature for 15 min before Sc(OTf)$_3$ (0.1 equiv) was added into the reaction mixture. The mixture was allowed to stir at room temperature and the reaction progress was checked by thin-layer chromatography (TLC). Once the limiting coupling partner was fully consumed, the reaction was quenched with triethylamine, and the mixture was directly loaded onto the silica gel under reduced pressure, which was further purified by silica gel column chromatography to afford the desired oligosaccharides or glycosides.

## Data availability

All data generated in this study are provided in the Supplementary Information and are available from the corresponding author upon request.

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

## Acknowledgements

We thank The National Research Foundation (NRF-CRP22-2019-0002), Ministry of Education (MOE-T2EP30120-0007), and A*STAR (A20E5c0087) from Singapore for the financial support and The National Natural Science Foundation of China (22207092), The Natural Science Basic Research Plan of Shaanxi Province of China (2022JQ-091), and The Fundamental Research Funds for the Central Universities (G2021KY05117) from China for financial support.

## Author contributions

X.-W.L., X.X., and H.D. conceived and directed the project. H.D., J.L., and X.-L.Z. designed and conducted the experiments. H.D., X.X., and X.-W.L. wrote the paper with input from all other authors.

## Competing interests

The authors declare no competing interests.
