## [Peer Review File · Nature Communications]

REVIEWER COMMENTS

Reviewer #1 (Remarks to the Author):

In this manuscript, Liu and co-workers developed a facile and efficient catalytic glycosylation which employed glycosyl ortho-2,2-dimethoxycarbonylcyclopropyl benzoates (CCBz) as donors promoted by non-toxic Sc(III) catalyst system. The reaction was featured by a new activation mode of glycosyl esters driven by the ring-strain release of intramolecularly incorporated donor-acceptor cyclopropane. The glycosyl CCBz donors enabled efficient construction of O-, S-, N-glycosidic linkages under mild conditions. Furthermore, the new glycosyl CCBz donors were used for the divergent synthesis of chitooligosaccharides. The manuscript is well-organized and well-written. This work provides a new glycosyl donor which could be a potential candidate for a new genre of catalytically activable glycosyl donors. In my opinion, this manuscript can be accepted for publication in this journal after some revisions.

- 1) Page 9, the compound numbers 2b-h should be as 1b-h.
- 2) Page 11 and Fig. 5, for the preparation of trisaccharide 11, compound 7 was used as glycosyl donor, not compound 8 was used. Please check and correct it.
- 3) For the synthesis of tetrasaccharide 15, the selenide donor 14 was used. How about the results if the glycosyl CCBz donor (the SePh leaving group in 14 was replaced by CCBz) was used?

Reviewer #2 (Remarks to the Author):

Carbohydrate chemistry has been one of the most exciting aspects of organic chemistry in the last few decades. Many new glycosylation methods have come into being, however, the tedious optimizations and trial-and-error procedures are still required for the synthesis of structurally diversified glycans. Thus, more ideal glycosylation methods are demanded to improve the synthesis efficiency and to speed up the development of glycoscience. In this reviewer's opinion, an ideal glycosylation method should have following properties: 1) the donor can be easily manufactured and has good stability, 2) the donor can be efficiently activated under a mild condition, 3) the glycosylation can proceed in an orthogonal manner.

This paper by Liu and co-workers reports an efficient catalytic glycosylation employing glycosyl ortho-2,2-dimethoxycarbonylcyclopropyl benzoates (CCBz) promoted by a readily accessible Sc(III)

catalyst system. The versatile three carbon building blocks are useful in organic synthesis due to both their reactivity and ease of preparation. The Lewis acid catalyzed ring-opening of donor-acceptor cyclopropanes (DAC) using nucleophiles is one of the straightforward methods for rapid access to 1,3-bifunctional compounds. In this paper, a DAC structure was introduced into the glycosyl ester donor to construct a new glycosyl donor (CCBz) with new activation mode mediated by non-covalent interactions. This rationally designed new glycosyl donor with an intramolecularly incorporated DAC featuring a dual-functional anchor: the metallophilic 1,3-dicarbonyl group as the activation site, and the ensuing enolate as an acid scavenger. Various glycosyl CCBzs were synthesized from the corresponding hemiacetals by carbodiimide-mediated esterification in good-to-excellent yields. This glycosylation reaction performed well under the catalyzation of Sc(OTf)₃ in various types of solvent. Acceptors ranged from aliphatic alcohols, sugar alcohols with primary, secondary and tertiary hydroxyl groups, benzoic acid derivative, phenol, sulfonamide and thiol were found to be compatible coupling partners in this glycosylation. A study on donor scope showed that glycosyl CCBzs derived from different monosaccharide- or oligosaccharide-based hemiacetals are competent coupling partners. Finally, this glycosylation method was applied successively to the synthesis of two chitooligosaccharide derivatives. Based on these results, this donor proved to be the prototype for developing next generation of catalytic glycosylation. The Supporting Information shows in detail the synthesis of compounds and documents the ¹H and ¹³C NMR spectra of all new compounds. Accordingly, the reported method can be reproduced.

Overall, this reviewer thinks that this article is worthy of publication in Nature Communications due to the novelty and potential of this glycosylation method. However, there are still a lot of spaces for authors to improve the paper. Notably, many writing mistakes that exist in the manuscript and Supporting Information showed that more careful attention should be paid to the preparation of a high-quality research paper. Thus, major alterations are required before an acceptance in Nature Communications might be considered (see "Specific points" below).

Specific points

1. The title "Catalytic Strain-Release Glycosylation" is too simple to accurately reflect the emphasis and content of the paper.
2. In the abstract, it's inappropriate to make the perspective like "With such, an array of peptidoglycan analogues could be prepared via the post-glycosylation modification strategy for novel antibiotic development to combat multidrug-resistant bacteria.". The discussion should focus on the developed method itself.
3. In the figure 1, "a, the chronology of catalytic glycosylation" should contain more commonly used donors such as thioglycoside.
4. In the figure 1d, it's inappropriate to claim that this work includes new bioactivities discovery.
5. The "Reaction optimization" section seems to be a combination of the reaction optimization and mechanistic investigation. It's suggested to separate them into two parts.

6. In the table 1 (or its revised version), final state (recovered, hydrolyzed, decomposed, or disappear) of the donor 1a in all glycosylation reactions is needed to show the selectivity of the activation.
7. Page 7, paragraph 3, line 10: "trance amount" should be corrected.
8. Page 8: please re-check the compound identifiers in the text.
9. Page 9, paragraph 2: please re-check the compound identifiers in the text.
10. Page 9, paragraph 2, line 6: "total 27 glycoarchitectures" should be corrected.
11. Page 10, paragraphs 1 and 2: this reviewer does not think it's meaningful to introduce so much elementary knowledge of the two target compounds in such a paper. But, instead, current status and problems of the chemical synthesis of two oligosaccharides should be intensively introduced to highlight the importance of this donor.
12. In the figure 5, please re-check the preparation of trisaccharide 11 through the assembly of two 1-O-benzylated glycosides.
13. In the figure 5, this reviewer is really curious to know usability of the CCBz donor in the synthesis of the tetrasaccharide 15 from the trisaccharide 13. A direct comparison between the CCBz donor and glycosyl selenide for such a challenging glycosylation is a good opportunity.
14. In the figure 5, it's unnecessary to provide an α/β ratio for the hemiacetal 16.
15. Please re-check the writing of EDC·HCl throughout this paper.
16. Page 11, paragraph 2, line 3: "Scheme 4" should be corrected.
17. Page 11, paragraph 1, line 6: "trisaccharide in 11" should be corrected.
18. Page 12, paragraph 1, line 10: "trance amount" should be corrected.
19. In the Supporting Information, the compound S2 should be identified as CCBzOH as written in the manuscript.
20. In the Supporting Information, please give some ¹³C chemical shifts to two digits after the decimal point to distinguish overlapping peaks.
21. In the Supporting Information, the "¹³C NMR (101 MHz, CDCl₃)" should be changed to "¹³C NMR (100 MHz, CDCl₃)".
22. In the Supporting Information, please assign all ¹H-NMR data for proper characterization of the new compounds.
23. In the Supporting Information, 2D NMR spectra of di-, tri-, and tetrasaccharides should be included.
24. In the Supporting Information, please re-check the structures of the compounds 2c and 3c. They are different to that in the manuscript.

25. In the Supporting Information, synthetic procedures of the reactions in the table 1 should be provided.
26. In the Supporting Information, page S27, please correct the compound name of 3cb.
27. In the Supporting Information, section 3, please re-check the compound identifiers in the text and pictures.
28. In the Supporting Information, "1H spectra and 13C spectra" should be changed to "1H and 13C spectra", "1H spectra" should be changed to "1H spectrum".

Reviewer #3 (Remarks to the Author):

The authors report a catalytic strain-release glycosylation employing glycosyl ortho-2,2-dimethoxycarbonylcyclopropyl benzoates (CCBz). The new glycosylation method enabled the efficient synthesis of an array of glycosides. However, the preparation of the leaving group CCBzOH required several steps according to the reference, which renders the synthesis of glycosyl CCBz a daunting task. The authors claimed that various O-, S-, N-glycosidic bonds were efficiently constructed, but only one example of S-glycoside and only one example of N-glycoside were shown in Fig. 3. Based on the CCBz strategy, the formal synthesis of TMG-chitotrimycin, Nod factor, Myc factor, and lipid IV could be established, although they contain the similar skeletons. The mechanism of the new glycosylation was proposed. Further characterization of the key intermediates could be necessary to elucidate the mechanism. In the past few years, a series of catalytic glycosylation methods have been developed. Nevertheless, only several glycosylation methods more than 15 years ago were listed in the chronology of catalytic glycosylation of Fig.1. The description of the catalytic glycosylation has not fully reflected the recent development in this field. Overall, this reviewer feels that the manuscript is not suitable to publish in high-profile journals such as Nature Communications.

Response to reviewer 1:

General comment: In this manuscript, Liu and co-workers developed a facile and efficient catalytic glycosylation which employed glycosyl *ortho*-2,2-dimethoxycarbonylcyclopropyl benzoates (CCBz) as donors promoted by non-toxic Sc(III) catalyst system. The reaction was featured by a new activation mode of glycosyl esters driven by the ring-strain release of intramolecularly incorporated donor-acceptor cyclopropane. The glycosyl CCBz donors enabled efficient construction of *O*-, *S*-, *N*-glycosidic linkages under mild conditions. Furthermore, the new glycosyl CCBz donors were used for the divergent synthesis of chitooligosaccharides. The manuscript is well-organized and well-written. This work provides a new glycosyl donor which could be a potential candidate for a new genre of catalytically activable glycosyl donors. In my opinion, this manuscript can be accepted for publication in this journal after some revisions.

Reply: We appreciate the reviewer's positive comments and strong support.

Comment 1: Page 9, the compound numbers **2b-h** should be as **1b-h**.

Reply: We apologize for the typos and we have made corrections accordingly. Please see the corrections on pages 8-9 in the revised manuscript.

Comment 2: Page 11 and Fig. 5, for the preparation of trisaccharide **11**, compound **7** was used as glycosyl donor, not compound **8** was used. Please check and correct it.

Reply: We thank the reviewer for pointing out these mistakes. The correct compound identifiers have been assigned to the corresponding compounds. For the corrected number series, please see revised Fig. 5 and revised Fig. 6 on page 11 and page 12, respectively.

Comment 3: For the synthesis of tetrasaccharide **15**, the selenide donor **14** was used. How about the results if the glycosyl CCBz donor (the SePh leaving group in **14** was replaced by CCBz) was used?

Reply: Thank you for your keen question. We have indeed considered employing a glycosyl CCBz donor to synthesize this tetrasaccharide, but we encountered some problems when synthesizing the donor. Specifically, the highly polar and acid-sensitive hemiacetal exhibited sluggish reactivity under normal coupling condition. However, in response to your and the second reviewer's interest in the synthesis of the tetrasaccharide using the glycosyl CCBz donor, we thus have revisited the synthesis of this donor and tried alternative synthesis approaches, and managed to prepare the *N*-2 Cbz protected glycosyl CCBz donor **15** by coupling the hemiacetal with *in-situ* generated CCBzCl (For experimental details, please refer to page S49 in revised supplementary information). When subjected to the strain-release glycosylation, the donor **15** gave corresponding tetrasaccharide in a slightly improved yield of 58% under the catalyzation of the Sc(III) catalyst. This result further demonstrates the effectiveness of the glycosyl CCBz donor in the synthesis of complex oligosaccharides.

To offer more useful information to the readers, sentences highlighted in yellow was added on page 11 in the revised manuscript, which read “**We also prepared glycosyl CCBz donor 15 from glycosyl selenide 14 in two steps to test the viability of glycosyl CCBz in the tetrasaccharide assembly (For the experimental details, please see SI). To our delight, this glycosylation reaction gave an improved yield of 58% under established condition. The enhanced reactivity and convenient operation further demonstrate the usability of glycosyl CCBz.**”

Again, we sincerely appreciate these valuable suggestions by the reviewer, which helped us to improve the quality of our paper significantly.

Response to reviewer 2:

General Comment: Carbohydrate chemistry has been one of the most exciting aspects of organic chemistry in the last few decades. Many new glycosylation methods have come into being, however, the tedious optimizations and trial-and-error procedures are still required for the synthesis of structurally diversified glycans. Thus, more ideal glycosylation methods are demanded to improve the synthesis efficiency and to speed up the development of glycoscience. In this reviewer’s opinion, an ideal glycosylation method should have following properties: 1) the donor can be easily manufactured and has good stability, 2) the donor can be efficiently activated under a mild condition, 3) the glycosylation can proceed in an orthogonal manner.

This paper by Liu and co-workers reports an efficient catalytic glycosylation employing glycosyl *ortho*-2,2-dimethoxycarbonylcyclopropyl benzoates (CCBz) promoted by a readily accessible Sc(III) catalyst system. The versatile three carbon building blocks are useful in organic synthesis due to both their reactivity and ease of preparation. The Lewis acid catalyzed ring-opening of donor-acceptor cyclopropanes (DAC) using nucleophiles is one of the straightforward methods for rapid access to 1,3-bifunctional compounds. In this paper, a DAC structure was introduced into the glycosyl ester donor to construct a new glycosyl donor (CCBz) with new activation mode mediated by non-covalent interactions. This rationally designed new glycosyl donor with an intramolecularly incorporated DAC featuring a dual-functional anchor: the metallophilic 1,3-dicarbonyl group as the activation site, and the ensuing enolate as an acid scavenger. Various glycosyl CCBzs were synthesized from the corresponding hemiacetals by carbodiimide-mediated esterification in good-to-excellent yields. This glycosylation reaction performed well under the catalyzation of Sc(OTf)₃ in various types of solvent. Acceptors ranged from aliphatic alcohols, sugar alcohols with primary, secondary and tertiary hydroxyl groups, benzoic acid derivative, phenol, sulfonamide and thiol were found to be compatible coupling partners in this glycosylation. A study on donor scope showed that glycosyl CCBzs derived from different monosaccharide- or oligosaccharide-based hemiacetals are competent coupling partners. Finally, this glycosylation method was applied successively to the synthesis of two chitooligosaccharide derivatives. Based on these results, this donor proved to be the prototype for developing next generation of catalytic glycosylation. The Supporting Information shows in detail the synthesis of compounds and documents the ¹H and ¹³C NMR spectra of all new compounds. Accordingly, the reported method can be reproduced.

Overall, this reviewer thinks that this article is worthy of publication in *Nature Communications* due to the novelty and potential of this glycosylation method. However, there are still a lot of spaces for authors to improve the paper. Notably, many writing mistakes that exist in the manuscript and Supporting Information showed that more careful attention should be paid to the preparation of a high-quality research paper. Thus, major alterations are required before an acceptance in *Nature Communications* might be considered (see "Specific points" below).

Reply: We sincerely thank the reviewer's comments on our paper. Your valuable comments shed light on finding new directions to further improve our glycosyl donor and develop more interesting glycosylation reactions.

We apologize for the typos and writing mistakes. After checking the manuscript again, the mistakes and typos, which are highlighted in red, have been corrected accordingly and listed here:

- 1) On page 1, the abstract part, "The salient features of our donor include the easy and scalable preparation of aglycon as well as the excellent thermostability, solubility and reactivity of the corresponding glycosyl CCBz in various organic solvent." has been corrected to "The salient features of our donor include the easy and scalable preparation of aglycon as well as the excellent thermostability, solubility and reactivity of the corresponding glycosyl CCBz in various organic solvents.";
- 2) On page 2, "glycosyl hetero-aromatic carboxylate esters" has been changed to "glycosyl heteroaromatic carboxylate esters";
- 3) On page 6, "A series of Lewis acid were initially tested in the presence of 5 Å molecular sieve (MS) in a 0.05 M solution of 1,2-dichloroethane (DCE) at room temperature for 2 h." has been corrected to "A series of Lewis acids were initially tested in the presence of 5 Å molecular sieve (MS) in a 0.05 M solution of **1a** in 1,2-dichloroethane (DCE) at room temperature for 2 to 5 h.";
- 4) On page 7, "revealing a orthogonal glycosyl ester type donors (entries 5-6)." has been corrected to "revealing a family of orthogonal glycosyl ester type donors (entries 5-6).";
- 5) On page 8, "Our facile synthesis of **3m** gave an example of catalytically feasible methods using cheap rare earth metal as the catalyst" has been corrected to "Our facile synthesis of **3m** gave an example of the catalytically feasible method using cheap rare earth metal as the catalyst";
- 6) On page 9, "L-rhamnosyl" has been corrected to "L-rhamnopyranosyl";
- 7) On page 11, "The global deprotection of benzyl groups and benzyloxycarbonyl (CBz) group by hydrogenolysis in a mixed solvent of iPrOH/THF/H₂O lead to the free tetrasaccharide **17** in 85% yield after purification." has been corrected to "The global deprotection of benzyl groups and benzyloxycarbonyl (CBz) group by hydrogenolysis in a mixed solvent of iPrOH/THF/H₂O led to the free tetrasaccharide **17** in 85% yield after purification.";
- 8) On page 12, "The tetrasaccharides obtained has several orthogonally modifiable sites, which could be selectively furnished to obtain a series of Lipid IV derivatives to test their anti-bacterial activities." Has been corrected to "Several orthogonally modifiable

sites on the tetrasaccharide could be selectively furnished to obtain a series of Lipid IV derivatives which hold the potential as bacterial TGases-targeting antimicrobial agents.”;

- 9) Besides, the supplementary information was thoroughly checked, and the typos and wrong compound identifiers and compound names have been corrected.

Comment 1: The title “Catalytic Strain-Release Glycosylation” is too simple to accurately reflect the emphasis and content of the paper.

Reply: We thank the reviewer’s valuable suggestion. After careful consideration, the prior title “Catalytic Strain-Release Glycosylation” has been amended to “Efficient and Versatile Formation of Glycosidic Bonds via Catalytic Strain-Release Glycosylation with Glycosyl ortho-2,2-Dimethoxycarbonylcyclopropyl Benzoate Donors”.

Comment 2: In the abstract, it’s inappropriate to make the perspective like “With such, an array of peptidoglycan analogues could be prepared via the post-glycosylation modification strategy for novel antibiotic development to combat multidrug-resistant bacteria.”. The discussion should focus on the developed method itself.

Reply: Thank you for your comment. The sentence has been removed in the revised manuscript.

Comment 3: In the figure 1, “a, the chronology of catalytic glycosylation” should contain more commonly used donors such as thioglycoside.

Reply: We thank the reviewer’s suggestion on Fig. 1. The purpose of Fig. 1a was to illustrate the trend of the catalytically activable glycosyl donors with increasing stabilities over time and emphasize the superiority of the glycosyl ester donors with excellent stability. The “catalytically activable glycosyl donor” herein is defined as a glycosyl donor which can be activated by a catalyst (including the ligand for TM catalysis) as the sole promoter. Although there are indeed several reports on the Bi(V) (Pohl N. L. *B. Angew. Chem. Int. Ed.* **52**, 8441-8445 (2013)) and Au(III) (Sureshan K. M. *Chem. Sci.* **7** 4259-4263 (2016)) catalyzed activation methods, the prevailing activation approaches of classical thioglycosides still require stoichiometric amount of thiophilic agents. Thus, we did not consider normal thioglycosides as meeting the criteria for our category of “catalytically activable glycosyl donors”. Meanwhile, some modifications of thioglycosides on their anomeric leaving group have been reported that enable their catalytic activations (For selected examples: please see (1) Zhu J. *ACS Catal.* **3**, 57-60 (2013); (2) Ragains J. R. *Org. Lett.* **20**, 5181-5185 (2018); (3) Ragains J. R. *Org. Lett.* **21**, 980-983 (2019); (3) Li M. *Eur. J. Org. Chem.* **2022**, e202101367, (2022); (4) Sun J.-S. *J. Am. Chem. Soc.* **145**, 3682-3695 (2023)). However, including all of these structures in Fig. 1a would likely confuse readers and distract from the main point of the figure, so we chose to omit these examples. Instead, to make the Fig. 1a clearer, the prior note “the chronology of catalytic glycosylation” has been changed to “the cornerstone glycosyl donors for catalytic

glycosylation in chronological order”. For the change, please see Fig. 1a on page 3 in the revised manuscript.

Meanwhile, to help the readers to understand our purpose of Fig. 1a and to provide more useful information, new contents were added, which are located on page 2, paragraph 2 in the revised manuscript and read **“A classical family of glycosyl donors with good stabilities is thioglycosides. Although multiple variations of thioglycoside have been developed to realize efficient catalytic glycosylation reactions,¹⁶⁻¹⁹ there is still a lot of space for the development of catalytic activation of classic thioglycoside. From a chronology with the cornerstone of catalytically activable glycosyl donors, we have identified a growing demand for ambiently stable donors that can be activated with catalytic amounts of promoters (Fig. 1). Inspired by these considerations, we endeavored to design a bench-top stable glycosyl donor that can be selectively activated by a specific catalyst in an orthogonal manner, utilizing a novel activation mode.”**

Comment 4: In the figure 1d, it's inappropriate to claim that this work includes new bioactivities discovery.

Reply: This inappropriate description in Fig. 1d has been removed according to your comment in the revised manuscript.

Comment 5: The “Reaction optimization” section seems to be a combination of the reaction optimization and mechanistic investigation. It's suggested to separate them into two parts.

Reply: In this suggestion, we did not perform the mechanistic investigation. There is only one control experiment to demonstrate the Sc(III)-catalyst was used to activate the DAC site instead of to activate the benzoate site.

To avoid possible confusion, the section title and table name on page 5 have been changed from **“Reaction optimization”** and **“Table 1. Reaction optimization”** to **“Reaction development”** and **“Table 1. Reaction development and control experiment”**, respectively.

Comment 6: In the table 1 (or its revised version), final state (recovered, hydrolyzed, decomposed, or disappear) of the donor **1a** in all glycosylation reactions is needed to show the selectivity of the activation.

Reply: In all cases of glycosylation with inferior yields, there is not any hydrolysis or decomposition of glycosyl CCBz donors. Thus the low yields could be attributed to the low conversion of the starting materials. To present more useful information, the reactions for this section were thus re-conducted to provide the readers with the yields of the recovered glycosyl donor (which can reflect the final state of the donor) and the yields of the departing product. All data have been added to the revised Table 1 on page 6 in the revised manuscript.

Comment 7: Page 7, paragraph 3, line 10: “trance amount” should be corrected.

Reply: The typo on page 7, paragraph 3, line 11 has been corrected now.

Comment 8: Page 8: please re-check the compound identifiers in the text.

Reply: The correct identifiers have been assigned to the corresponding compounds.

Comment 9: Page 9, paragraph 2: please re-check the compound identifiers in the text.

Reply: The identifiers for these compounds have been corrected.

Comment 10: Page 9, paragraph 2, line 6: “total 27 glycoarchitectures” should be corrected.

Reply: The “total 27 glycoarchitectures” on page 9, paragraph 2, line 6 has been corrected to “27 oligosaccharides and glycosides”.

Comment 11: Page 10, paragraphs 1 and 2: this reviewer does not think it’s meaningful to introduce so much elementary knowledge of the two target compounds in such a paper. But, instead, current status and problems of the chemical synthesis of two oligosaccharides should be intensively introduced to highlight the importance of this donor.

Reply: We greatly thank the reviewer’s advice on the content, which is important for improving our work. Following your suggestion, this part has been altered by streamlining the content on the biological functions of chitooligosaccharides. Additionally, the advancements in current synthetic methods of chitooligosaccharides and related references were added. For details, please see page 10, paragraphs 2 and 3.

Comment 12: In the figure 5, please re-check the preparation of trisaccharide 11 through the assembly of two 1-O-benzylated glycosides.

Reply: The compound identifiers have been corrected according to your comment. The trisaccharide was synthesized by the glycosylation reactions between disaccharide acceptor **10** and glycosyl CCBz **7**. Please see the revised Fig. 5 on page 11 in the revised manuscript.

Comment 13: In the figure 5, this reviewer is really curious to know usability of the CCBz donor in the synthesis of the tetrasaccharide **15** from the trisaccharide **13**. A direct comparison between the CCBz donor and glycosyl selenide for such a challenging glycosylation is a good opportunity.

Reply: We sincerely appreciate the reviewer’s suggestion. As per your and the first

reviewer's advice, the CCBz donor **15** was successfully prepared. Employing this donor, the tetrasaccharide **16** was obtained in a slightly improved yield of 58% at room temperature under milder reaction condition with simpler operation, which demonstrated the usefulness of glycosyl CCBz donor in the assembly of complex oligosaccharide. We also addressed a similar question raised by the first reviewer. For more details, please also refer to the additional contents highlighted in yellow on page 11 in the revised manuscript and our response to the comment 3 raised by the first reviewer.

Comment 14: In the figure 5, it's unnecessary to provide an α/β ratio for the hemiacetal **16**.

Reply: The anomeric ratio for this hemiacetal has been removed from the figure. Please refer to the corrected Fig. 5 on page 11 in the revised manuscript.

Comment 15: Please re-check the writing of EDC·HCl throughout this paper.

Reply: The compound name for this chemical has been generalized to "EDC·HCl".

Comment 16: Page 11, paragraph 2, line 3: "Scheme 4" should be corrected.

Reply: "Scheme 4" has been changed to "Fig. 6". The correction can be found on page 12 in the revised manuscript.

Comment 17: Page 11, paragraph 1, line 6: "trisaccharide in **11**" should be corrected.

Reply: We thank the reviewer for pointing out this mistake. "trisaccharide in 11 over two steps" on page 11, line 7 has been corrected to "trisaccharide 11 in two steps".

Comment 18: Page 12, paragraph 1, line 10: "trance amount" should be corrected.

Reply: The typo on page 12, line 18 has been corrected.

Comment 19: In the Supporting Information, the compound **S2** should be identified as **CCBzOH** as written in the manuscript.

Reply: The compound has been generalized to "**CCBzOH**" throughout the revised manuscript and supplementary information.

Comment 20: In the Supporting Information, please give some ^{13}C chemical shifts to two digits after the decimal point to distinguish overlapping peaks.

Reply: We apologize for this mistake. All signals of ^{13}C NMR spectra with two digits for overlapped peaks are given in the revised supplementary information.

Comment 21: In the Supporting Information, the “¹³C NMR (101 MHz, CDCl₃)” should be changed to “¹³C NMR (100 MHz, CDCl₃)”.

Reply: We have made corrections on the ¹³C NMR spectra accordingly. The frequency is corrected to 100 MHz in the revised supplementary information.

Comment 22: In the Supporting Information, please assign all ¹H-NMR data for proper characterization of the new compounds.

Reply: The significant signals on the ¹H NMR spectra for all new compounds have been assigned in sections 3-5 in the supplementary information.

Comment 23: In the Supporting Information, 2D NMR spectra of di-, tri-, and tetrasaccharides should be included.

Reply: We thank the reviewer’s suggestion. We have performed 2D NMR experiments for compounds **11**, **13**, **17**, **21**, **22** and **24**. The COSY and HSQC spectra for these compounds have been attached in the supplementary information on pages S116, S118, S123, S126, S128 and S131.

Comment 24: In the Supporting Information, please re-check the structures of the compounds **2c** and **3c**. They are different to that in the manuscript.

Reply: We apologize for this mistake. Now the structures in both manuscript and supplementary information have been re-checked and corrected. Please refer to page 8 in the revised manuscript and page S21 in the revised supplementary information for the correct structures, respectively.

Comment 25: In the Supporting Information, synthetic procedures of the reactions in the table 1 should be provided.

Reply: A new section titled “Reaction development of strain-release glycosylation” has been added in supplementary information, which provides the general synthetic procedure for the reaction development and the data for compounds **3a** and **4**. For more details, please refer to “Section 2. Reaction development of strain-release glycosylation” on page S18 in revised supplementary information.

Comment 26: In the Supporting Information, page S27, please correct the compound name of **3cb**.

Reply: We apologize for the mistake. The compound name for this compound has been corrected. Meanwhile, the names for other compounds have been re-checked and corrected.

Comment 27: In the Supporting Information, section 3, please re-check the compound identifiers in the text and pictures.

Reply: We apologize for this mistake. The compound identifiers in supplementary information have been re-checked and corrected.

Comment 28: In the Supporting Information, “¹H spectra and ¹³C spectra” should be changed to “¹H and ¹³C spectra”, “¹H spectra” should be changed to “¹H spectrum”.

Reply: We have corrected the mistakes accordingly in the revised supplementary information.

Finally, thank you very much again for helping us to polish our paper.

Response to reviewer 3:

General Comment: The authors report a catalytic strain-release glycosylation employing glycosyl *ortho*-2,2-dimethoxycarbonylcyclopropyl benzoates (CCBz). The new glycosylation method enabled the efficient synthesis of an array of glycosides.

Reply: First of all, we sincerely appreciate the insightful comments and suggestions by this reviewer. Your valuable comments and suggestions are very important to the revisions of our manuscript.

Comment 1: However, the preparation of the leaving group CCBzOH required several steps according to the reference, which renders the synthesis of glycosyl CCBz a daunting task.

Reply: We fully understand your valid concern. Indeed, it takes 5 steps to have the leaving group CCBzOH prepared from starting materials commercially available. However, we consider the preparation process feasible for the following reasons.

- (1) The starting materials for the synthesis of CCBzOH, i.e. dimethyl malonate and *o*-tolualdehyde, are very cheap and robust chemicals. Additionally, all other reagents required for the synthesis are similarly easy to deal with, and none of them need special cautions for moisture and air sensitivity. Moreover, this route involves no transition metal and thus is considered environmentally friendly;
- (2) The reactions involved in the synthesis are well-established, robust, and high-yielding. All reactions are air tolerant and except for the cyclopropylation step, all reactions are also moisture tolerant with apparent scalability. A working synthetic chemist can easily reproduce the synthesis on a scale of 20 grams in less than three days to attain CCBzOH effortlessly;
- (3) The work-up and purification procedures for each step are also very convenient, and each step affords the product with a high yield over 85%. Separation of the product from starting materials is easy since their polarities are significantly different. For the oxidation of the aldehyde to the acid, the purification is even not necessary. The

CCBzOH could be obtained with over 95% purity by simply washing the resulting solid with hexane;

- (4) The utilization of CCBzOH as the efficient agent to achieve the activation of other unactivated hydroxyl group are still in progress in our lab. It is believed that this reagent will finally be accepted by the broad chemistry community for its versatility.

We again thank the reviewer to raise the concern about the preparation of CCBzOH. To ensure that all of the reactions involved in the synthesis of CCBzOH can be reproduced by anyone who is interested in our glycosylation reactions, **the detailed synthetic procedures for step-by-step decagram-scale preparation of CCBzOH with our tips and tricks are added in the supplementary information to ensure the reproducibility.** For the modified section for the synthesis of CCBzOH, please see section 1, the synthesis of CCBzOH on page S7 in the revised supplementary information.

Comment 2: The authors claimed that various *O*-, *S*-, *N*-glycosidic bonds were efficiently constructed, but only one example of *S*-glycoside and only one example of *N*-glycoside were shown in Fig. 3.

Reply: We would like to thank this reviewer for the valuable comments, which have helped us to improve the manuscript. As per the reviewer's suggestion, we have removed the word "various" from the abstract to make our expression more precise.

In order to further demonstrate the versatility of our glycosyl CCBz donor in *S*-glycosylation, two additional types of *S*-nucleophiles, aliphatic thiol 1-octanethiol **2r** and sugar-derived thiol 2,3,4,6-tetra-*O*-benzoyl-1-thio- β -D-galactopyranose **2s** were investigated. The coupling reaction with **2r** proceeded efficiently to provide thioglycoside **3r** in a yield of 64%. However, no reaction occurred when **2s** was used as the acceptor. While this result was less satisfactory, it is believed that there is still a lot more for us to explore, and the reviewer's insightful suggestions have enlightened us about the new possibilities for our strain-release glycosylation. Specifically, we are now exploring the strain-release glycosylation of glycothiols, which promises access to the 1,1'-*S*-linked sugar derivatives of biological and medicinal interests.

With respect to *N*-glycosylation, we are actively working on using the glycosyl CCBz donor to achieve the glycosylation reactions of other *N*-nucleophiles like nucleobases and asparagine derivatives. Since these *N*-nucleophiles usually have weaker nucleophilicity, the synthesis of *N*-glycosides is considered challenging. To further showcase the usefulness of glycosyl CCBz in the preparation of nucleosides and asparagine glycosides, we would like to publish these results in a separate paper.

According to the reviewer's comment, additional lines highlighted in yellow have been added on page 9 in the revised manuscript, which read "**Finally, the *S*-acceptors were also viable for the construction of *S*-linked glycosides, as exemplified by the efficient coupling reactions of donor **1a** with aromatic thiol **2q** and aliphatic thiol **2r**, respectively. It should be noted that in all cases of successful glycosylation reactions employing heteroatom nucleophiles, the direct ring-opening of DAC moiety by acceptors was not observed, denoting the intramolecular cyclization can be kinetically favorable even in the presence of a heteroatom nucleophile. However,**

it was disappointing that when galactose-derived anomeric thiol 2s was subjected to our optimal condition, no reaction occurred. Currently, the reason for the unpleasant consequence was unknown and we are still working on the synthesis of 1,1'-thiosaccharide congeners by strain-release glycosylation because of their important roles as drug candidates.⁶⁵⁻⁶⁷”.

Comment 3: Based on the CCBz strategy, the formal synthesis of TMG-chitotrimycin, Nod factor, Myc factor, and Lipid IV could be established, although they contain the similar skeletons.

Reply: At first glance, the structures for these COS derivatives indeed appear similar; however, the strategies we deployed to synthesize chitooligosaccharides and Lipid IV tetrasaccharide vary in many aspects when testing the viability of glycosyl CCBz as glycosylating agents in the synthesis of complex oligosaccharides, each with its unique emphasis and purpose.

For the chitooligosaccharide synthesis, an iterative glycosylation strategy was used to test the ability of strain-release glycosylation in a linear synthetic route. Meanwhile, because the TMG-chitotrimycin, Nod factor and Myc factor share the same tetrasaccharide skeleton except for the substitution patterns in the terminal monosaccharide, the main consideration was the protection fashion of the 2-amino group of the terminal glycosyl donor. In the initial manuscript, a selenoglycoside was used for the synthesis of tetrasaccharide, which might undercut the novelty of the work. However, by successfully synthesizing *N*-2 Cbz protected glycosyl CCBz donor and exposing it to the strain-release glycosylation, the challenging assembly of tetrasaccharide **16** was achieved in an improved yield of 58% under milder condition with cheap and abundant Sc(III) catalyst instead of a stoichiometric amount of promoter, demonstrating the usability of the glycosyl CCBz as an efficient glycosylating agent in the assembly of complex oligosaccharides.

When it comes to the Lipid IV tetrasaccharide synthesis, a convergent strategy was selected to test the ability of oligosaccharide glycosyl CCBz donor for complex oligosaccharide synthesis. As has been demonstrated in the Fig. 6, the operation at the anomeric position and the glycosylation stage of disaccharide glycosyl CCBz are smooth to furnish the tetrasaccharide on a gram-scale. Meanwhile, because the peptidoglycan is not confined as the simple disaccharide or tetrasaccharide, the potential for the precise preparation of bigger oligomers of peptidoglycans was also considered by leaving an orthogonally cleavable TBS ether at the O-4 in the non-reducing terminal unit. Besides, in the overall synthesis, thioglycoside was used as the acceptor twice while the aglycon transfer side-reaction was almost suppressed. Considering the aglycon transfer is one of the most common side-reaction when the thioglycoside is involved in the glycosylation reaction as the acceptor (For one recent example cited in our paper, please see ref. 94, Xiao, G. *Chem. Sci.* **12**, 5143-5151 (2021)), our method offered a more flexible choice of acceptor over other glycosylation reactions.

Comment 4: The mechanism of the new glycosylation was proposed. Further characterization of the key intermediates could be necessary to elucidate the mechanism.

Reply: We appreciate your valuable suggestions. Identification of the key intermediate by variable temperature NMR will indeed help to shed light on the mechanism and we had tried conducting VT NMR experiments. However, to our disappointment, we did not observe any change in the spectrum during the temperature elevation from -78 °C to room temperature, probably due to the poor solubility of Sc(OTf)₃ in the NMR tube. It is speculated that the chelation of Sc(III) catalyst with DAC could increase its solubility and trigger the cyclization and departure stage. It is worth noting that during the review stage of this paper, we published another paper on the DAC-mediated thioglycoside activation in *CCS chem* (CCS chem. 10.31635/ccschem.023.202202671 (2023)), in which we performed some preliminary mechanistic studies. Considering the successful separation of compound **4** and the mechanistic studies in our published work, we believe that the mechanism proposed in this manuscript is plausible. This paper is cited as ref. 57 in the revised manuscript to provide readers with more information.

We are currently trying our best to explore an appropriate activation system to conduct the VT NMR studies to characterize the key intermediate during the reaction process, as suggested by the reviewer. This study promises to provide further corroboration to our proposed mechanism.

Comment 5: In the past few years, a series of catalytic glycosylation methods have been developed. Nevertheless, only several glycosylation methods more than 15 years ago were listed in the chronology of catalytic glycosylation of Fig.1. The description of the catalytic glycosylation has not fully reflected the recent development in this field.

Reply: We appreciate the reviewer's feedback regarding our inappropriate description in Fig. 1a. The purpose of this figure is meant to briefly review the important catalytically activable glycosyl donor, as explained in our response to the comment 3 of the second reviewer. To clarify its purpose, the Fig. 1a has thus been renamed to **"the cornerstone glycosyl donors for catalytic glycosylation in chronological order."** The new description was also added on page 2, paragraph 2, which reads **"A classical family of glycosyl donors with good stabilities is thioglycosides. Although multiple variations of thioglycoside have been developed to realize efficient catalytic glycosylation reactions,¹⁶⁻¹⁹ there is still a lot of space for the development of catalytic activation of classic thioglycoside. From a chronology with the cornerstone of catalytically activable glycosyl donors, we have identified a growing demand for ambiently stable donors that can be activated with catalytic amounts of promoters (Fig. 1). Inspired by these considerations, we endeavored to design a bench-top stable glycosyl donor that can be selectively activated by a specific catalyst in an orthogonal manner, utilizing a novel activation mode."**

In addition, we acknowledged that we missed some important references to reflect the detailed development of catalytic glycosylation reactions in a more complete manner. Thus, the following changes are made to provide more information to experts in this field:

- (1) The initial first paragraph has been divided into two sections, and a description of the advances in catalytic activation of thioglycoside congeners has been added in the new

- second paragraph on page 2, paragraph 2;
- (2) Glycosyl allenoate and new recent reference for glycosyl heteroaromatic carboxylate esters have been added on page 2, paragraph 3 in the revised manuscript;
 - (3) Recent advances in efficient O- and N-glycosylation using alkynyl-based glycosyl donors have been added on page 3, paragraph 2, which read ***“Inspired by this elegant activation mode, a series of catalytic glycosylation reactions have been developed recently for efficient formation of O-³²⁻³⁵ and N-glycosidic linkages.³⁶⁻³⁸”***,
 - (4) The advancements in catalytic glycosylation using cyclopropane-fused glycosyl donors as glycosylating agents for the synthesis of unnatural glycosides have been added on page 4, paragraph 1, which read ***“In addition, while catalytic glycosylation reactions involving cyclopropane-fused glycosyl donors have successfully demonstrated excellent control of anomeric selectivity to produce unnatural glycosides,⁵⁵⁻⁵⁶ an effective approach to obtain natural sugar derivatives via catalytic strain-release-driven glycosylation remains elusive.”***.

We would like to thank this reviewer once again for giving insightful suggestions for our paper.

REVIEWERS' COMMENTS

Reviewer #1 (Remarks to the Author):

The authors fully addressed my concerns. This revised manuscript can be accepted for publication in this journal.

Reviewer #2 (Remarks to the Author):

The authors have carefully addressed the concerns of the reviewers. The paper is now much better. Thus, I recommend acceptance of the manuscript for publication in Nature Communications after two issues are addressed (see "Specific points" below).

Specific points

1. Page 9, paragraph 2, line 6: "27 oligosaccharides and glycosides" should be changed to "21 oligosaccharides and glycosides".
2. For the compound 1a in the Table 1, and the compound 15 in the Figure 5, the " $\alpha:\beta$ " is not shown.

To the comments from reviewer 1:

General Comments: The authors fully addressed my concerns. This revised manuscript can be accepted for publication in this journal.

Reply: We thank the reviewer's recommendation.

To the comments from reviewer 2:

General Comments: The authors have carefully addressed the concerns of the reviewers. The paper is now much better. Thus, I recommend acceptance of the manuscript for publication in Nature Communications after two issues are addressed (see "Specific points" below).

Reply: We thank the reviewer's recommendation.

Comment 1: Page 9, paragraph 2, line 6: "27 oligosaccharides and glycosides" should be changed to "21 oligosaccharides and glycosides".

Reply: The "27 oligosaccharides and glycosides" has been corrected to "21 oligosaccharides and glycosides", which is highlighted in yellow on page 6 in revised manuscript.

Comment 2: For the compound 1a in the Table 1, and the compound 15 in the Figure 5, the " $\alpha:\beta$ " is not shown.

Reply: The anomeric ratios for the two compounds have been added in Table 1 and Figure 5 on page 17 and page 22, respectively, in revised manuscript.